# Inhibition of human-HPV hybrid ecDNA enhancers reduces oncogene expression and tumor growth in oropharyngeal cancer

Takuya Nakagawa [1,2,3] ✉, Jens Luebeck [4], Kaiyuan Zhu[4], Joshua T. Lange[5], Roman Sasik [6], Chad Phillips[1], Sayed Sadat[1], Sara Javadzadeh[4], Qian Yang[7], Abdula Monther[1], Santiago Fassardi[1], Allen Wang [7], Kersi Pestonjamasp[8], Brin Rosenthal[6], Kathleen M. Fisch [6], Paul Mischel [5], Vineet Bafna [4] & Joseph A. Califano [1,9] ✉

Extrachromosomal circular DNA (ecDNA) has been found in most types of human cancers, and ecDNA incorporating viral genomes has recently been described, specifically in human papillomavirus (HPV)-mediated oropharyngeal cancer (OPC). However, the molecular mechanisms of human-viral hybrid ecDNA (hybrid ecDNA) for carcinogenesis remains elusive. We characterize the epigenetic status of hybrid ecDNA using HPVOPC cell lines and patient-derived tumor xenografts, identifying HPV oncogenes E6/E7 in hybrid ecDNA are flanked by previously unrecognized somatic DNA enhancers and HPV L1 enhancers, with strong cis-interactions. Targeting of these enhancers by clustered regularly interspaced short palindromic repeats interference or hybrid ecDNA by bromodomain and extra-terminal inhibitor reduces E6/E7 expression, and significantly inhibits in vitro and/or in vivo growth only in ecDNA(+) models. HPV DNA in hybrid ecDNA structures are associated with previously unrecognized somatic and HPV enhancers in hybrid ecDNA that drive HPV ongogene expression and carcinogenesis, and can be targeted with ecDNA disrupting therapeutics.

Extrachromosomal circular DNA (ecDNA) represents a frequent driver of focal oncogene amplification during carcinogenesis, and the subsequent high expression of the genes carried on ecDNA[1–3]. This phenomenon is facilitated by the circular structure and open-chromatin configuration of ecDNA[4], allowing transcriptional factors to easily access ecDNA. The unique spatial reorganization of ecDNA is thought to create additional topological domains, providing opportunities for previously unrecognized interactions between oncogenes and distant

regulatory elements, and ecDNA mediated *cis*-interactions between enhancers and oncogenes ("enhancer hijacking"). Oncogenes residing on chromosomal DNA in the human genome may also be activated using so-called "mobile enhancers" from free-floating ecDNA[5]. Furthermore, ecDNA segments in close proximity may also interact through "ecDNA hubs"[6,7].

Recent studies have shown that in Human papillomavirus (HPV)-mediated cancers, human DNA and viral genomes can form circular

[1]Moores Cancer Center, University of California, San Diego, La Jolla, CA, USA. [2]Department of Otorhinolaryngology, Head and Neck Surgery, Chiba University, Graduate School of Medicine, Chiba, Japan. [3]Health and Disease Omics Center, Chiba University, Chiba, Japan. [4]Department of Computer Science and Engineering, UC San Diego, La Jolla, CA, USA. [5]Department of Pathology, Stanford University School of Medicine, Stanford, CA, USA. [6]Center for Computational Biology and Bioinformatics, University of California, San Diego, La Jolla, CA, USA. [7]Department of Cellular and Molecular Medicine, Center for Epigenomics, University of California, San Diego, La Jolla, CA, USA. [8]Cancer Center Microscopy Core, University of California, San Diego, La Jolla, CA, USA. [9]Department of Otolaryngology—Head and Neck Surgery and Gleiberman Head and Neck Cancer Center, University of California, San Diego, La Jolla, CA, USA. ✉e-mail: tanakagawa@health.ucsd.edu; tnakagawa@chiba-u.jp; jcalifano@health.ucsd.edu

human–viral hybrid ecDNA (hybrid ecDNA). HPV is an approximately 8kbp circular double-stranded DNA virus and includes oncogenes called E6 and E7[8,9]. HPV-mediated oropharyngeal cancer (HPVOPC) incidence has dramatically increased over the last 2 decades and has become one of the fastest growing cause of solid organ cancer death in the US[10,11]. In our previous analysis, hybrid ecDNA was identified in 16 out of 56 cases (around 30%) of HPVOPC[12]. While somatic integration of HPV oncogenes into the human genome is known as a carcinogenic driver of HPVOPC[13,14], the molecular mechanisms by which hybrid ecDNA drives carcinogenesis have not been well described. Understanding these mechanisms and potential therapeutic vulnerabilities in hybrid ecDNA(+) HPVOPC are an important, unmet challenge in understanding and treating HPVOPC.

We hypothesized that in addition to oncogene amplification, HPV oncogene transcriptional upregulation via *cis*-interactions in hybrid ecDNA between enhancers and HPV oncogenes, and proximity of hybrid ecDNA segments themselves are deeply involved in mediating the carcinogenic molecular functions of HPVOPC.

Here, we investigated the chromatin status of hybrid ecDNA using HPVOPC cell lines and PDX tumors and identified previously unrecognized active enhancers in hybrid ecDNA. The HPV oncogenes E6/E7 inside hybrid ecDNA were flanked by previously unrecognized somatic enhancers in addition to HPV enhancers within the L1 region. HiC-seq identified a strong interaction between these enhancers and E6/E7, supporting cis-interactions inside hybrid ecDNA. Inactivation of this enhancer by clustered regularly interspaced short palindromic repeats (CRISPR) interference reduced the expression of E6/E7 and proliferation. Furthermore, bromodomain and extra-terminal (BET) inhibitor treatment targeting hybrid ecDNA structures in ecDNA(+) tumors

resulted in the significant inhibition of tumor growth in both cell lines and PDX tumors not seen in hybrid ecDNA(−) tumors, suggesting avenues for therapeutic intervention in HPVOPC.

## Results
### Identification of hybrid ecDNA in HPVOPC cell lines and PDX tumors

To determine hybrid ecDNA status, we performed WGS and RNA-seq on 2 HPVOPC cell lines, HMS001 and SCC154, and 2 patient-derived xenograft (PDX) tumors from clinical tumor samples of HPVOPC patients, described here as PDX_A and PDX_C. As negative control, we used the cell-line NOKSI, (spontaneously immortalized human cell line derived from normal oral mucosa that is HPV negative and known to not carry ecDNA). Analysis of WGS and RNA-seq data showed the presence of hybrid DNA molecules and hybrid transcripts in HMS001 and PDX_A but not in PDX_C, SCC154 or NOKSI. Using amplicon architect (AA), hybrid ecDNA was detected in HMS001 and PDX_A, but not in SCC154, PDX_B or NOKSI (Fig. 1A–D). Across samples, each sequence of hybrid ecDNA was unique, consistent with our previous analyses in clinical samples of HPVOPC[12]. Long read DNA-seq also detected structural variants for both viral and human segments on the same sequenced molecule, consistent with the AA results (Supplementary Fig. 1A and B). Furthermore, the sequence alignments of the HMS001, or PDX_A reads to the HPV16 genome found no reads that suggested a structure in which the two endpoints of HPV16 used for integration into the human genome had become fused together without involving human DNA, suggesting that the absence of an episome containing only HPV DNA (Supplementary Fig. 1C and D). Reanalysis using previously published long read DNA-seq of SCC154

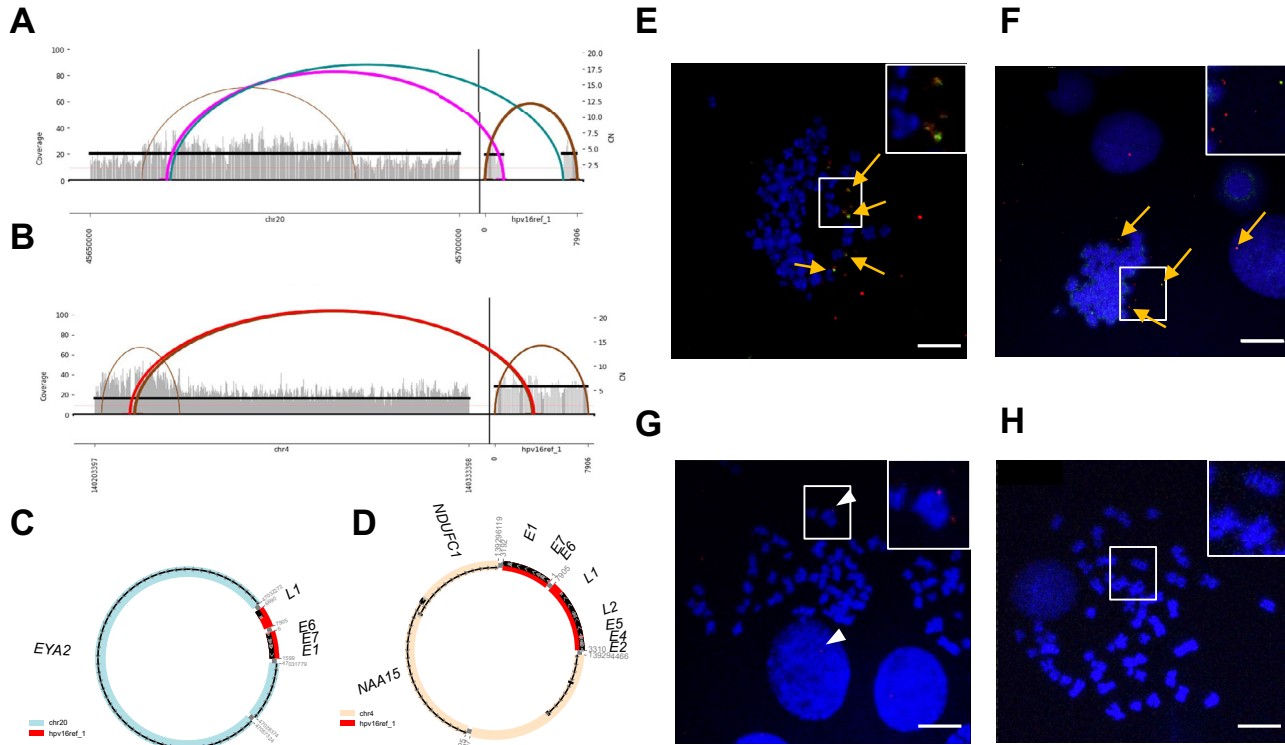

**Fig. 1 | Identification of hybrid ecDNA in HPVOPC using AA and FastViFI and validation of hybrid ecDNA by multi-FISH.** Examples of circular hybrid ecDNA suggested by AA results in HMS001 (**A** and **C**) and PDX tumor PDX_A (**B** and **D**). Multi-FISH using each ecDNA specific probe (green) and HPV specific probe (red) for metaphase spread cells showed overlapping of each probe signal in the same place only in hybrid ecDNA(+) samples (allow in **E** and **F**). Hybrid ecDNA(−) cell line

SCC154 only showed the red integrated HPV signal in chromosome (arrowhead in **G**). Control cell line NOKSI did not show any signal (**H**). Cell nuclei were counterstained with DAPI (blue). Scale bar shows 10 μm. An expanded view of white box in each is shown upper right corner. Each experiment was at least repeated twice with similar results (**E**–**H**).

only showed integrated HPV genome, also suggesting the absence of an episome containing only HPV DNA. For the cytogenetic validation of each hybrid ecDNA, multi-DNA FISH targeting each human genome and HPV genome on hybrid ecDNA was performed using a metaphase-spread in cell line and short-time cultured PDX tumors. Overlapping of somatic and HPV probes was located outside of condensed chromosomes in HMS001 (hybrid ecDNA(+)) and PDX_A, confirming the existence of hybrid ecDNA (Fig. 1E and F), but no such overlapping probes were found in SCC154 (hybrid ecDNA(−); Fig. 1G and H). We also observed integrated HPV-only signals on human chromosome only in SCC154, suggesting the existence of integrated HPV DNA in SCC154[15] (Fig. 1G). The mean copy numbers of HPV/cell in HMS001, PDX_A, SCC154, and NOKSI were 4.14, 4.11, 2.3, and 0, respectively (Supplementary Table 1).

## Previously unrecognized enhancers are created in hybrid ecDNA structures

To elucidate chromatin status of hybrid ecDNA, we performed ChIP-seq for H3K27ac (activation mark), H3K4me1 (enhancer mark), H3K4me3 (promoter mark), and ATAC-seq on HPVOPC cell lines and PDX tumors. Contrary to our expectations, previously described somatic enhancer regions were not hijacked, but clusters of H3K4me1 and H3K27ac peaks, indicative of active enhancers, were found in the regions with hybrid ecDNA (Fig. 2A). Intriguingly, a previously unrecognized cluster of H3K4me1 and H3K27ac peaks was observed in HMS001, but not in SCC154 or NOKSI, suggesting the creation of a enhancer element in the hybrid ecDNA (Fig. 2A). Strikingly, the HPV integration site within hybrid ecDNA exists in the center of these enhancers in hybrid ecDNA. Specifically, H3K4me1 and H3K27ac enrichment was noted in the HPV L1 region within the hybrid ecDNA enhancer as part of a larger enhancer region and ATAC-seq confirmed these enhancers exist in open chromatin regions (Fig. 2B). In addition, the E6/E7 promoter was surrounded by these enhancers indicating an enhance-promoter complex. The same pattern was observed within the hybrid ecDNA region in PDX_A (Fig. 2C and D). Furthermore, the transcription of the human sequence in each hybrid ecDNA was quite elevated compared to a different cell line without hybrid ecDNA at that same location (Fig. 2B and D). Although the somatic genomic regions associated with each hybrid ecDNA were quite different and unique between HMS001 and PDX_A, we found that in each case, the E6/E7 promoter was surrounded by previously unrecognized enhancers in somatic regions and an enhancer in the HPV L1 region. Based on these data, we hypothesized that HPV gene expression, specifically E6/E7, was upregulated by previously unrecognized somatic human and HPV hybrid enhancer complexes on hybrid ecDNA. These phenomena only occurred in hybrid ecDNA samples, suggesting a hybrid ecDNA(−) specific mechanism.

## Human and viral genomes on hybrid ecDNA interact directly with each other

To elucidate enhancer and HPV DNA interactions in hybrid ecDNA, HiC-seq was performed using the same cell lines and PDX tumors. Human somatic genomic sequences on hybrid ecDNA were divided into 2 segments (S1 and S2). The S1 enhancer region significantly interacted with the HPV L1 region and the S2 enhancer region significantly interacted with HPV E6/E7 regions in HMS001 ($P < 2 \times 10^{-7}$, $P < 4 \times 10^{-4}$, respectively) (Fig. 3A and B). On the other hand, there was no such interaction in SCC154 that lacked hybrid ecDNA (Fig. 3C). This phenomenon was confirmed in PDX tumors as well (Fig. 3D–F). In PDX_A, the enhancer existed mainly in the S1 segment, and the S1 enhancer region significantly interacted with HPV L1 and E6/E7 regions ($P < 2 \times 10^{-3}$, $P < 2 \times 10^{-3}$, respectively) (Fig. 3D and E), and there was no such interaction in hybrid ecDNA(−) PDX_C (Fig. 3F). Although each hybrid ecDNA structure was unique, each of the somatic enhancer regions closely interacted with HPV genome,

confirming the direct interaction of the human and viral genomes in hybrid ecDNA.

## CRISPR interference targeting of enhancers on hybrid ecDNA blocks HPV oncogene expression

To elucidate the functional role of previously unrecognized enhancers on hybrid ecDNA, we performed CRISPR interference (CRISPRi), targeting specific hybrid ecDNA enhancers in HMS001. We generated dCas9-KRAB stable cell lines; HMS001, SCC154, and NOKSI. gRNAs targeting S1: the long part (gRNA#1) and the S2: the short part (gRNA#2) of the enhancer on hybrid ecDNA of HMS001 and the non-target control were used (Fig. 4A). We confirmed the expression of dCas9 after doxycycline induction (Fig. 4B and C, Supplementary Fig. 2A and B). Consistent with our hypothesis, E6 and E7 expression were specifically reduced by the repression of the S2 enhancer region on hybrid ecDNA by CRISPRi (Fig. 4D). This phenomenon was not seen in SCC154 or NOKSI without hybrid ecDNA (Supplementary Fig. 2C and D), supporting the notion that the enhancer on hybrid ecDNA was uniquely created. Furthermore, contrary to the nontarget control or S1 repression, S2 repression significantly inhibited the proliferation only in HMS001 ($P = 0.004$) (Fig. 4E–G). This indicates that previously unrecognized enhancer regions in hybrid ecDNA specifically drive the expression of HPV oncogenes.

## Hybrid ecDNAs are physically associated and can be targeted therapeutically in vitro and in vivo

Next, we investigated the potential for HPVOPC therapeutic strategies targeting hybrid ecDNA. We tested a Bromo- and Extra-Terminal domain (BET) inhibitor (JQ1) as a potential therapeutic agent to target BRD4 as a key linker of ecDNA[6]. We hypothesized that disruption of the direct interaction of the enhancer in the somatic genome that interacts with HPV E6/E7 on hybrid ecDNA would inhibit HPV oncogene expression and reduce downstream gene targeting and proliferation.

To investigate the spatial properties of hybrid ecDNA in the cells, we performed multi-probe FISH using an EYA2 probe and an HPV probe in super-resolution with and without JQ1 treatment for HMS001 cells. FISH signals for hybrid ecDNA tended to accumulate in the nucleus, and these signals were reduced after JQ1 treatment. This suggests that the copy number of hybrid ecDNA molecules decreased after JQ1 treatment (Fig. 5A and B).

To investigate gene expression changes after JQ1 treatment, we first checked E6/E7 expression after JQ1 treatment (Fig. 6A). E6/7 expression was only downregulated in the hybrid ecDNA(+) HMS001 cell line—not in hybrid ecDNA(−) SCC154. MYC is one of the key downstream oncogenes regulated by BRD4, and as expected, MYC was downregulated by JQ1 even in hybrid ecDNA(−) cell lines. (Fig. 6B, C, and Supplementary Fig. 3). In addition, upon JQ1 treatment, E6/E7 expression was reduced in a concentration-dependent manner (100 nM and 1 μM) at 6 h after JQ1 treatment and more rapidly at 24 h (Fig. 6B and C). A similar result was seen in proliferation assays in HMS001 and SCC154, in which JQ1 treatment significantly inhibited tumor growth only in hybrid ecDNA(+) HPVOPC, but not in ecDNA(−) HPVOPC ($P = 0.03$, $P = 0.12$, respectively) (Fig. 6D). Comparison of ChIP-seq data between JQ1 treatment and DMSO in HMS001 indicated that while we found many BRD4 peaks significantly reduced after JQ1 treatment, we found many H3K27ac peaks significantly upregulated after JQ1 treatment (Supplementary Fig. 4A and B). To clarify specific pathways affected by chromatin alterations, 810 differentially upregulated H3K27ac peaks along with H3K4me1 peaks (FC > 2 and lfdr<0.3) were identified. Gene Ontology analysis showed that GO terms reported being suppressed by HPV infection were restored, such as, "Apoptosis" and "epithelial cell differentiation" (Supplementary Fig. 4C). Comparison of RNA-seq data between JQ1 treatment and DMSO in HMS001 also included "positive regulation of programmed

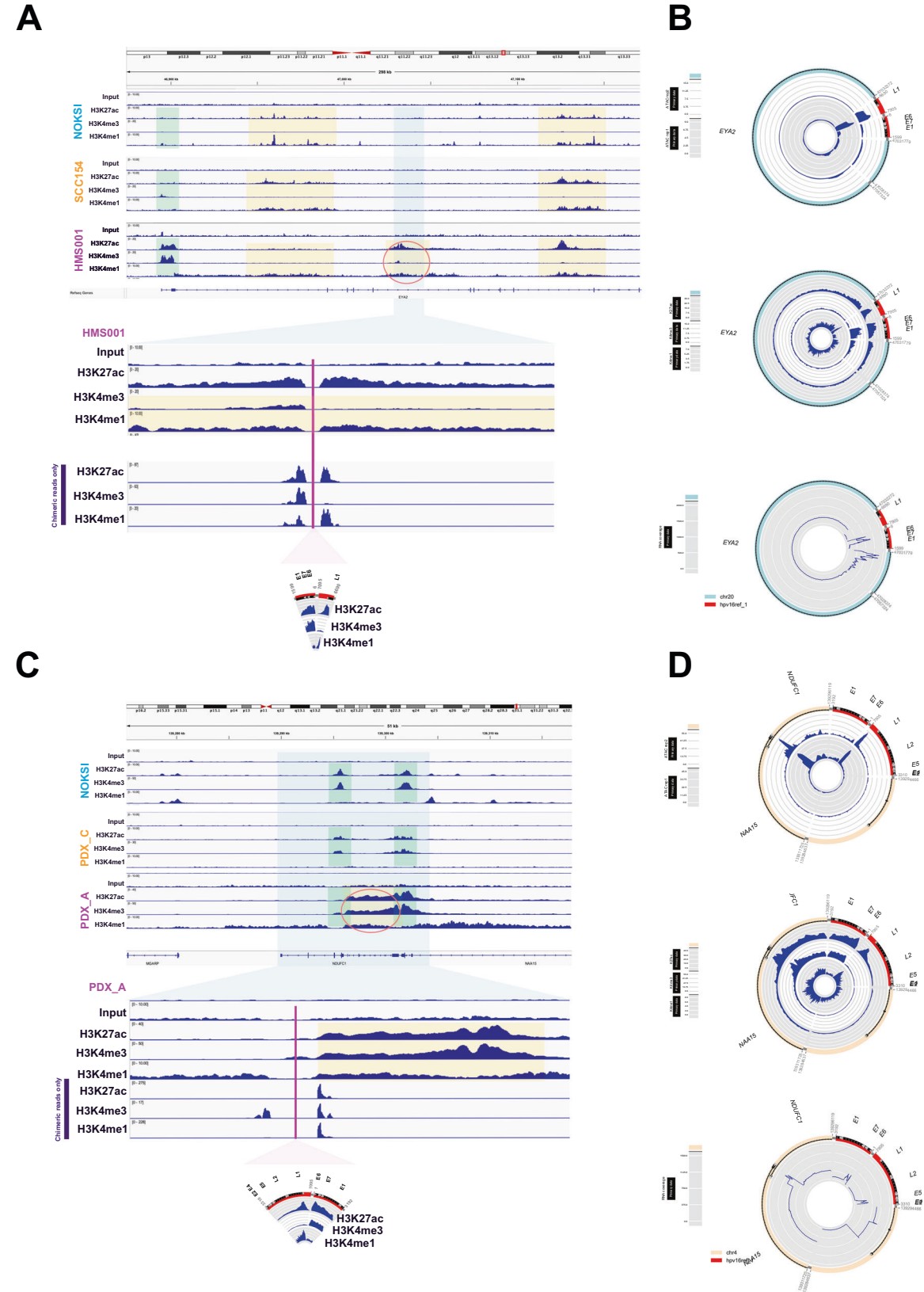

cell death" and "p53 transcriptional gene network" in the GO term for gene groups with significantly increased expression (Supplementary Fig. 4D and E). GO terms for significantly downregulated genes including "cell cycle" are also supportive data for the suppression of HPV. On the other hand, HiC-seq showed no significant changes after the JQ1 treatment, suggesting that the structure of the hybrid ecDNA

itself is not affected by the JQ1 treatment (Supplementary Fig. 4F and G).

We investigated JQ1 treatment in an in vivo model using a patient-derived xenograft model of HPVOPC (Fig. 7A) with hybrid ecDNA(+) (PDX_A) and hybrid ecDNA(−) (PDX_C and PDX004). In the PDX_A JQ1 treatment group, tumor growth was significantly inhibited compared

**Fig. 2 | Detecting active enhancers using ChIP-seq and identifying HPV integration mechanisms in hybrid ecDNA.** ChIP-seq results of Input, K4me3 (promoter), K4me1 (enhancer), and K27ac (activation mark) for NOKSI (normal control), SCC154 (hybrid ecDNA(−) HPVOPC) and HMS001 (hybrid ecDNA(+) HPVOPC) were shown. Promoter was indicated by the light green box, enhancers by the yellow box, and the sequences included in hybrid ecDNA of HMS001 by the light blue box. The active enhancer was marked by the red oval (*top*). Expanded hybrid ecDNA sequence with only chimeric reads of each ChIP-seq data were shown, and HPV integration occurred in the exact center of the active enhancer mark (pink line) (*bottom*) (**A**). Hybrid ecDNA in HMS001with ATAC-seq (*top*), ChIP-seq (*middle*), and RNA-seq (*bottom*) were shown in the CycleViz plot (**B**). The same analysis using NOKSI (normal control), PDX_C (hybrid ecDNA(−)HPVOPC), and PDX_A (hybrid ecDNA(+) HPVOPC) were shown. The active enhancer that did not exist in other cell lines made a complex with 2 promoters (*bottom*) (**C** and **D**).

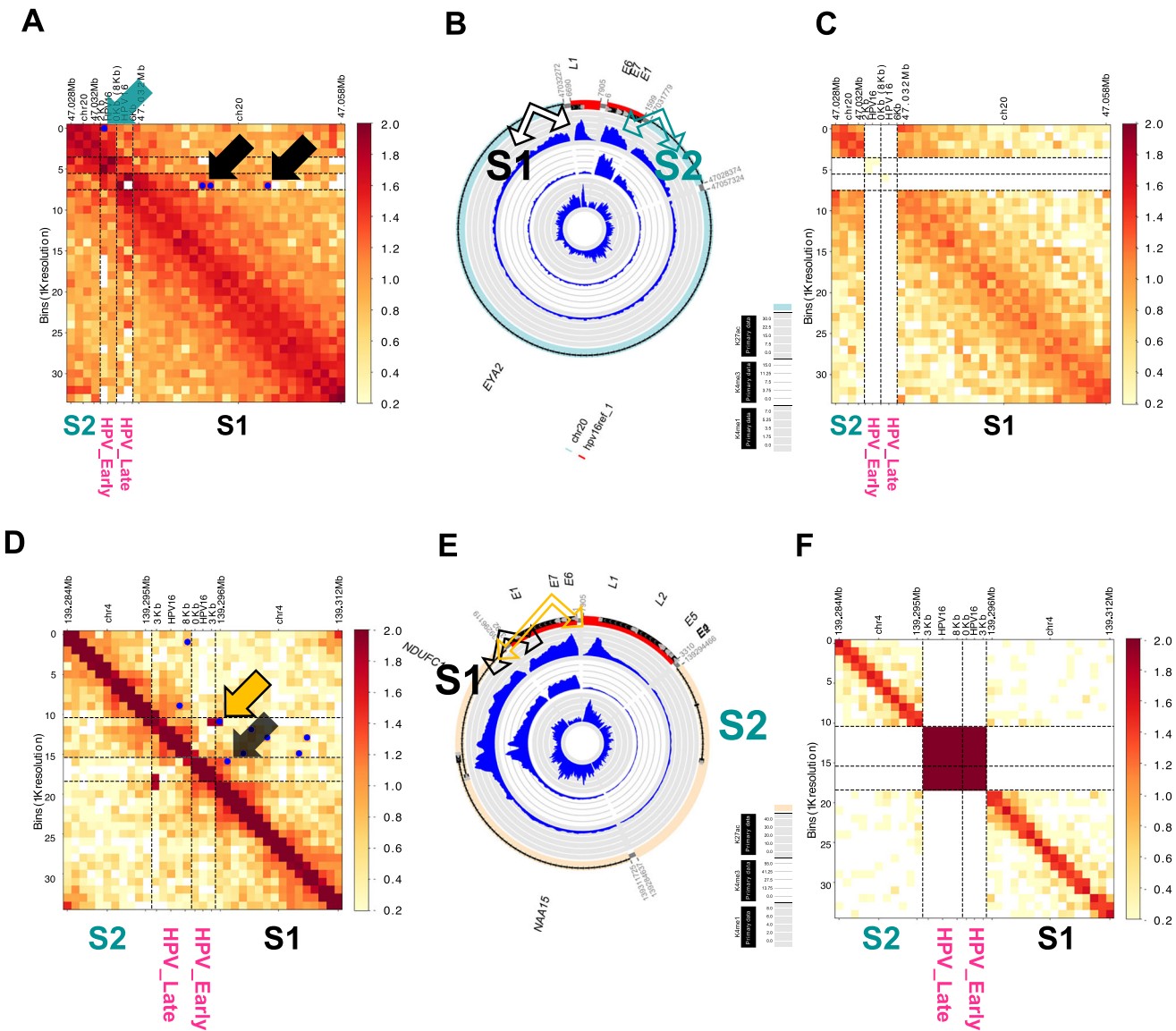

**Fig. 3 | Human and viral genomes on hybrid ecDNA interacted directly with each other.** *Cis*-interactions between enhancer and HPV in hybrid ecDNA were analyzed by HiC-seq. Human genome regions on hybrid ecDNA were divided into 2 segments (S1 and S2). The S1 enhancer region interacted with the HPV L1 region (black arrow in **A** and **B**), and the S2 enhancer region significantly interacted with the HPV E6/E7 regions in HMS001 (Right green arrow in **A** and **B**). Blue dots indicate significant interactions. On the other hand, there was no such interaction in SCC154 that lacked hybrid ecDNA (**C**). This phenomenon was confirmed in PDX tumors (**D**–**F**). In PDX_A, the enhancer existed only in the S1 segment, and the S1 enhancer region significantly interacted with the HPV L1 and E6/E7 regions in PDX_A (black and yellow arrows in **D** and **E**). Blue dots indicate significant interactions. On the other hand, there was no such interaction in PDX_C that lacked hybrid ecDNA (**F**). Although each hybrid ecDNA structure was unique, each of the human enhancer regions closely interacted with HPV, confirming the direct interaction of the human and viral genomes on the hybrid ecDNAs.

to the vehicle control group (tumor volume: $P = 2 \times 10^{-5}$, tumor weight: $P = 1 \times 10^{-4}$, respectively) (Fig. 7B–E). Consistent with our hypothesis, E6/E7 expression was also reduced after JQ1 treatment in this model ($P < 1 \times 10^{-4}$) (Fig. 7F). On the other hand, in PDX_C and PDX004: hybrid ecDNA(−) models, JQ1 did not significantly inhibit tumor growth

(Fig. 7G–L), suggesting specific targeting of hybrid ecDNA(+) tumors. Furthermore, multi-FISH targeting hybrid ecDNA using a short-time culture cell line from PDX_A tumors showed a reduction of hybrid ecDNA FISH signals after JQ1 treatment (Supplementary Fig. 5A–C). HiC-seq showed no significant changes after JQ1 treatment in PDX A as

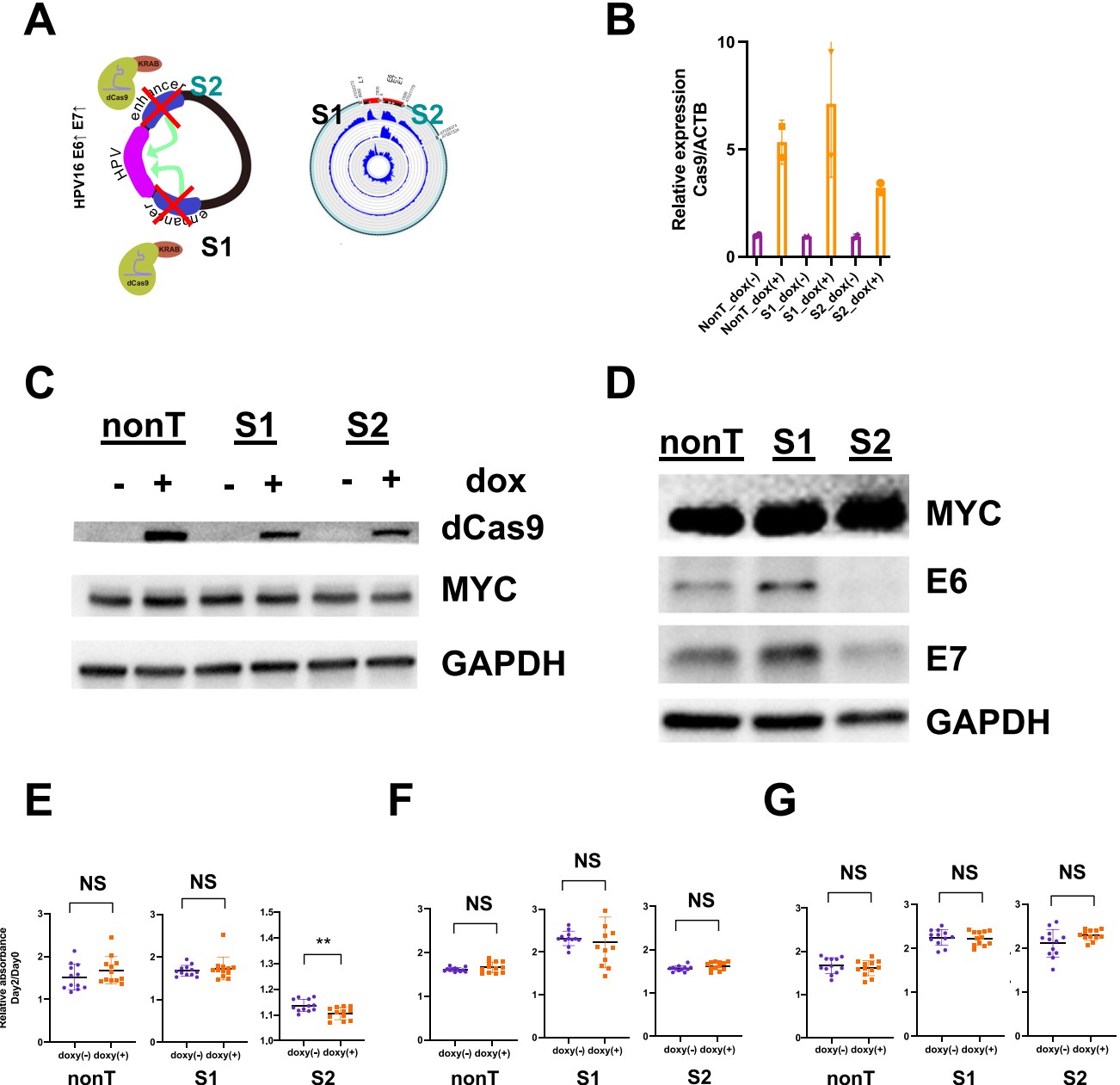

**Fig. 4 | CRISPR interference targeting enhancers on hybrid ecDNA blocks HPV oncogene expression.** CRISPR interference, using dCas9-KRAB to target enhancers on the hybrid ecDNA of HMS001, was performed. gRNAs targeting S1: the long part (gRNA#1) and S2: the short part (gRNA#2) of the enhancer on hybrid ecDNA of HMS001 and nontarget controls were used (**A**). The expression of dCas9 after doxycycline induction was confirmed by qPCR (**B**) and Western blotting (**C**). Two biological replicates were used and the median with SD was shown in qPCR results (B). MYC and GAPDH were used as controls (C). Western blotting results of E6 and

E7 of each CRISPRi condition indicate E6 and E7 expression were reduced by CRISPRi targeting the S2 enhancer (**D**). Proliferation assay results targeting the nontarget control (nonT), S1 enhancer, and S2 enhancer in HMS001, SCC154, and NOKSI were shown. Twelve biological replicates were used and median with SD were shown. Dox induction significantly inhibited the proliferation only in targeting S2 in HMS001 (**P = 0.004, two-tailed student's t-test) (**E**), but not in SCC154 or NOKSI (**F** and **G**, two-tailed student's t-test). Source Data is available for panels **B** and **E**–**G**.

well as in HMS001 (Supplementary Fig. 5D and E). Taken together, these data suggest the potential for specific therapy targeting hybrid ecDNA through interruption of HPV-mediated gene expression changes.

## Discussion

While the formation of ecDNA is commonly associated with DNA damage, such as chromosome shattering (chromothripsis), and breakage-fusion-bridge cycles (BFBs)[2,3,16,17], viral integration into the human genome also induces genomic instability[18–21] and causes the formation of hybrid ecDNA and carcinogenesis. Although HPV

integration often occurs in non-coding regions and some hot spots as previously reported, the location of integration is thought to be random[22,23]. Furthermore, HPV integration is reported to change chromatin accessibility status and activate surrounding genes[24,25], but the mechanisms behind this phenomenon are yet to be elucidated. Although HPV integration around originally existing enhancers and hijacking of known enhancers has been reported previously[26], we have observed that HPV integration creates previously unrecognized enhancers in ecDNA that did not exist in OPC samples without HPV integration at that locus. We found HPV genome integration sites were surrounded by previously unrecognized enhancers and these regions

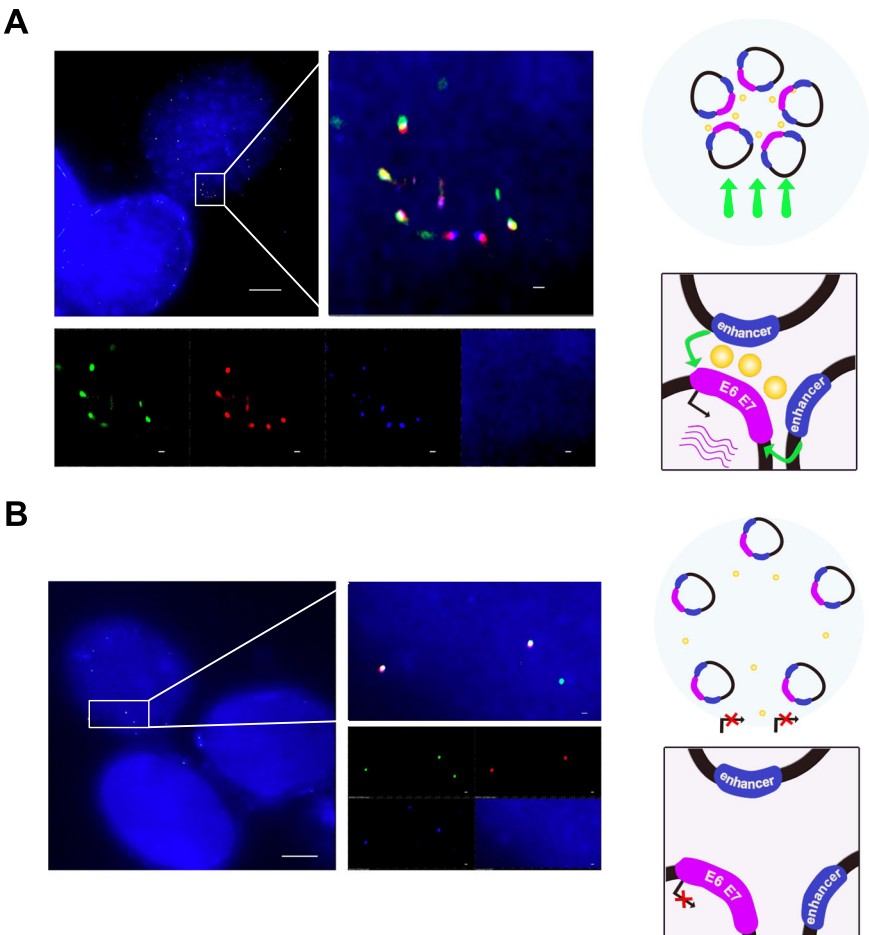

**Fig. 5 | Hybrid ecDNAs interference with each other and reduced after JQ1 treatment.** Multi-probe FISH, using an EYA2 probe and an HPV probe on hybrid ecDNA on super-resolution DMSO and JQ1 treatment of HMS001 cells, is shown (**A** and **B**). Hybrid ecDNAs were observed nearby in the nucleus in the "no treatment" condition (**A**). Each signal (green: EYA2, red: HPV, and blue: DAPI) was also shown separately (*bottom*) (**A**). FISH signals of hybrid ecDNAs were reduced after JQ1 treatment (**B**). Scale bar shows 5 µm (0.2 µm in the expanded picture). Cartoon illustrating how disruption of interferences between hybrid ecDNAs may decrease transcription is illustrated (*right* in **A** and **B**). Each experiment was repeated twice with similar results.

formed hybrid ecDNA in both HPVOPC cell lines and PDX tumors from clinical HPVOPC patients. In addition, we also demonstrated the HPV genome itself can serve as an enhancer region within ecDNA. This is perhaps one reason why HPV integration may be seen as "random" or not biased to integration with known enhancer/promoter regions: HPV genome sequences themselves can serve as enhancers and can induce enhancer activity in genome regions that do not serve as enhancers in the normal physiologic state.

In our study, cis-interactions between HPV and previously unrecognized enhancers on hybrid ecDNA were confirmed by HiC-seq, and CRISPRi targeting this enhancer reduced the expression of E6/E7 and proliferation consistently. Interestingly, in HMS001, HPV was surrounded on both sides by previously unrecognized enhancers, but only one enhancer (S2) showed inhibition of HPV E6/E7 expression and cell growth. This is consistent with the interaction in HiC-seq data, which also showed a strong interaction between S2 and E6/E7 in HMS001and may be related to high expression of S2 by RNA-seq. On the other hand, the S1 and L1 enhancers form one large enhancer together, and repression of the S1 enhancer alone by CRISPRi was not sufficient to repress E6/E7 expression, because the L1 enhancer was still active. We also made the intriguing observation that HPV genome regions, including the L1 region, can serve as enhancers when integrated into hybrid ecDNA structures. Detailed examination of the hybrid ecDNA structure along with the ChIP-seq data shows that an enhancer of HPV in a cell line (HMS001) or PDX models, forms a complex with an enhancer in the L1 region of HPV. This is consistent with the fact that the suppression of HPV E6/E7 expression did not occur by repression of the enhancer on the human side alone. Although the cell lines and PDX from different patients had completely different hybrid ecDNAs, they shared the composition of previously unrecognized enhancers as well as HPV L1 enhancers flanking E6/E7, suggesting that L1 regions might be key enhancer regions driving HPV expression in hybrid structures.

Furthermore, we explored the role of interference between hybrid ecDNAs in HPVOPC. Although it was difficult to specifically define 'hybrid ecDNA hubs' in our analysis, hybrid ecDNAs were found near each other in the nucleus, and HPV E6/E7 expression was strikingly reduced after JQ1 treatment. It is reasonable that the copy number of HPV in hybrid ecDNA is not as high as the copy number of the onco-gene in other cancer types[6], because hybrid ecDNA can drive high levels of HPV transcription by coexisting and interfering with each other.

Of note, our study has limitations. Due to the modest sample size of cell lines and experimental systems, the results bear additional confirmation and exploration with additional systems and models.

In the past few years, the development of therapies that target ecDNA has been accelerating[27]. In this study, we identified that targeting the enhancer on hybrid ecDNA by CRISPRi reduced the expression of E6/E7 and proliferation in vitro model in HPVOPC. Fur-thermore, we showed that specific targeting of hybrid ecDNA with a

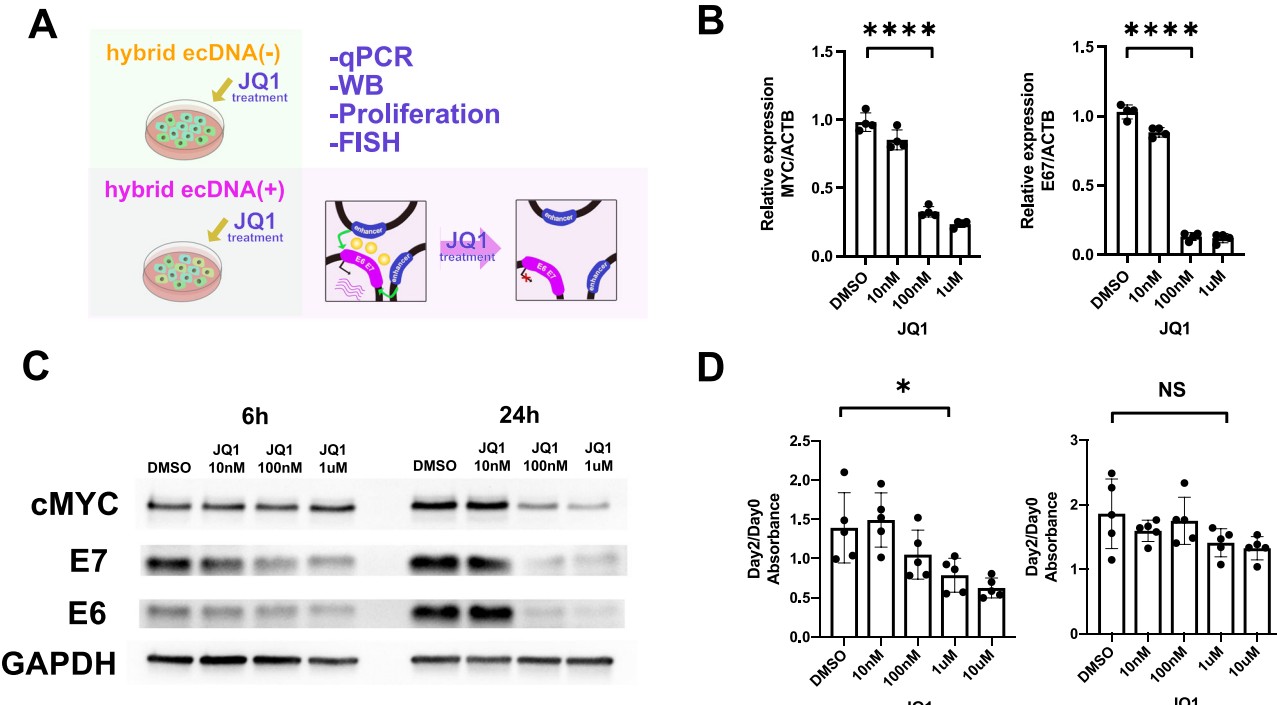

**Fig. 6 | JQ1 treatment on hybrid ecDNA significantly blocks HPV oncogene expression and proliferation.** JQ1 treatment of HMS001 and SCC154 was performed. The schema of the experiments was shown (**A**). qPCR results of MYC and E6/E7 using HMS001 were shown. ACTB was used for internal control. Four biological replicates were used. MYC and E6/E7 expressions were reduced in a JQ1 concentration-dependent manner in qPCR using HMS001 (****$P < 1 \times 10^{-4}$, ****$P < 1 \times 10^{-4}$, respectively, two-tailed Student's $t$-test) at 24 h. Error bars represent SD (**B**). MYC and E6/E7 expressions were reduced in a JQ1 concentration-dependent manner in western blotting using HMS001 at 6 and 24 h after JQ1 treatment (**C**). Proliferation assay using JQ1 for HMS001(*left*) and SCC154 (*right*) were shown. Five biological replicates were used and median with SD were shown. JQ1 treatment significantly inhibited tumor growth only in HMS001 in 1uM, but not in SCC154 (*$P = 0.03$, $P = 0.12$, respectively, two-tailed student's $t$-test) (D). Source Data is available for panels **B** and **D**.

BET inhibitor in HPVOPC was able to effectively reduce tumor growth and oncogene expression in both in vivo and in vitro models. In the context of these preclinical models targeting hybrid ecDNA, BET inhibitors are a potential targeted therapy for patients with hybrid ecDNA(+) tumors. This provides a rationale for biomarker-driven targeting of hybrid ecDNA(+) HPVOPC, and investigation of other types of treatment for ecDNA(+) HPVOPC that specifically targets ecDNA.

## Methods

### Clinical samples
Clinical specimens for whole-genome sequencing (WGS), long read DNA-sequencing, ChIP-seq, ATAC-seq, HiC-seq, and RNA-seq were collected from HPVOPC patients who underwent surgery at Jacobs Medical Center at the University of California, San Diego (UCSD) Health. Written informed consent was obtained from all patients. These samples were shared with the UCSD Human Research Protection Program (institutional review board (IRB)-approved protocol HRPP# 181755) by way of the Moores Cancer Center Biorepository and Tissue Technology resource. Two independent pathologists confirmed that the purity of the primary tumor was at least 80%. HPV status was determined by p16 immunohistochemistry or in-situ hybridization.

### Patient-derived xenografts (PDX)
Female 4-week-old nonobese diabetic/severe combined immunodeficiency (NOD/SCID) mice were purchased from The Jackson Laboratory. Fresh clinical HPVOPC tumor samples (PDX_A: 55-year-old male, PDX_C: 76-year-old male, and PDX004: 75-year-old male) were extracted at surgery and cut into approximately 3 mm pieces and transplanted into 6–8-week-old female NOD/SCID mice. Xenografting procedures were described previously[28,29]. The animal study protocol

(ASP, #S16200) was approved by the University of California San Diego (UCSD) Institutional Animal Care and Use Committee (IACUC) and complies with the ethical code for animal experiments. The ASP criteria for maximum tumor size are a maximum dimension greater than 15 mm or the presence of an ulcer. All mice were euthanized according to ASP guidelines. The maximum tumor burden allowed by the ASP was not exceeded. The laboratory mice were housed in a facility maintained at 18–23 °C with 40–60% humidity and a 12-h light/dark cycle. All personnel adhered to strict protective measures, including wearing scrubs or gowns, masks, hairnets, dedicated footwear, and disposable gloves while in the vivarium.

### Cell culture
SCC154 was purchased from the American Type Culture Collection (ATCC: CRL-3241). HMS001 was a gift from the James Rocco lab (Ohio State). NOKSI was a gift from the lab of Silvio Gutkind. SCC154 and NOKSI were grown in DMEM (Sigma-Aldrich, St. Louis, MO). HMS001 was grown in 2/3 of DMEM and 1/3 of F12 Medium (Sigma-Aldrich). All media were supplemented with 10% FBS (Sigma-Aldrich), 1% penicillin/streptomycin (Sigma-Aldrich), and plasmocin (InvivoGen, Toulouse, France). Cells were cultured at 37 °C with 5% $CO_2$.

For short-time cell culture of PDX tumors, PDX mice were euthanized for tissue retrieval, and tumors were dissected. Then, tumors were digested and isolated for a short-time cell culture. Procedures were described elsewhere[30]. Cells were grown in defined keratinocyte serum-free media (Invitrogen, Carlsbad, CA) supplemented with 1% antibiotics, 5 ng/ml mouse epidermal growth factor (Invitrogen), and $2 \times 10^{-11}$ M cholera (Sigma-Aldrich) at 37 °C with 5% $CO_2$. Mycoplasma testing was conducted monthly using the MycoAlert Plus Mycoplasma Detection Kit (Lonza, Basel, Switzerland) and all cell lines were confirmed negative.

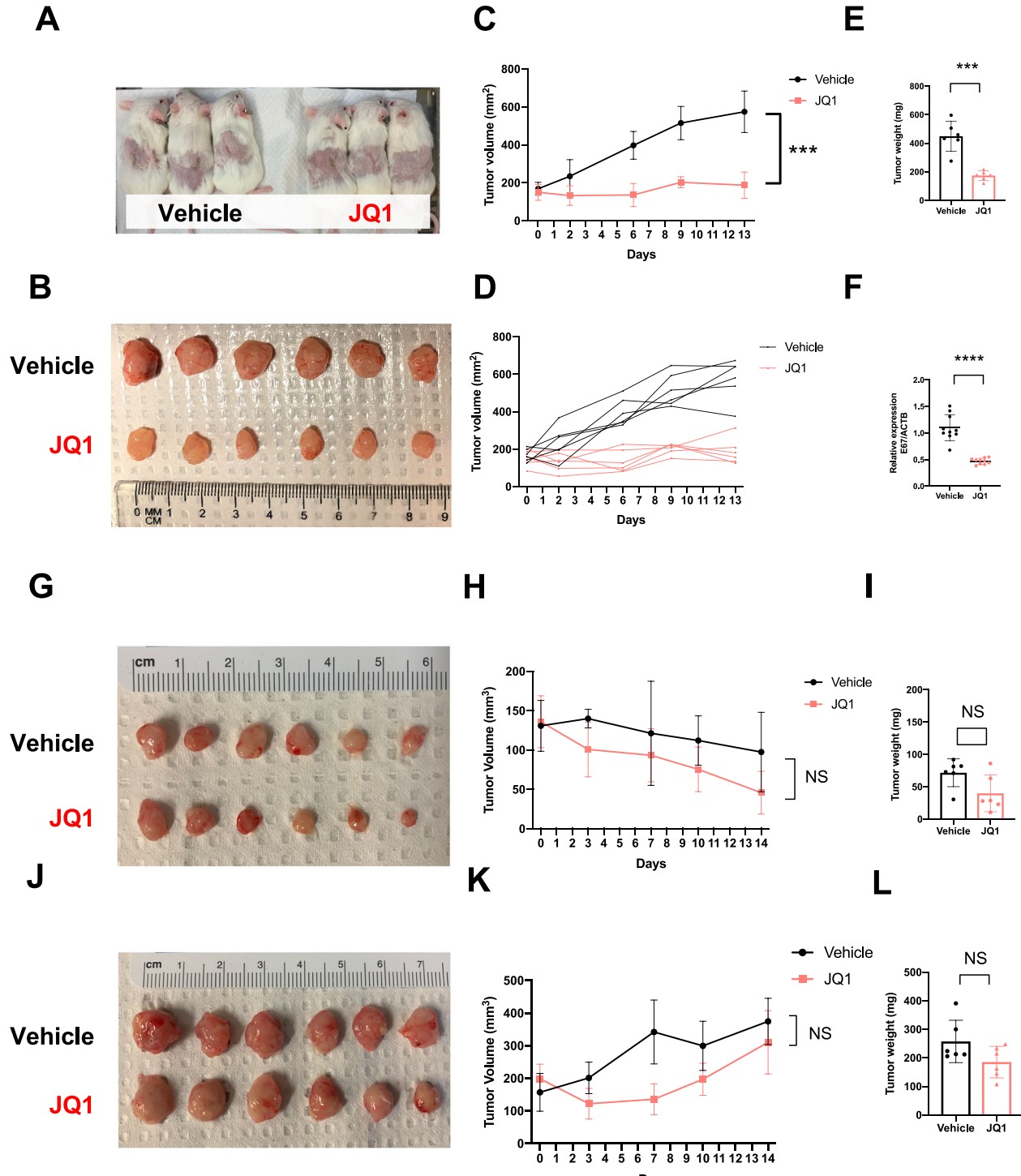

**Fig. 7 | JQ1 inhibited proliferation only in hybrid ecDNA(+) HPVOPC PDX tumors.** JQ1 treatment was performed in HPVOPC PDX models. Six PDX_A (hybrid ecDNA+) mice were divided into a vehicle control group and a JQ1 treatment group. Each mouse possessed tumors in both flanks (**A**). Tumors were harvested after 2 weeks of vehicle control or JQ1 treatment (**B**). Tumors were harvested after 2 weeks of JQ1 treatment or vehicle control. Tumor volumes of each condition are shown. Six replicates for each group were used. Median with SD (**C**) and each replicate (**D**) were shown. In the PDX_A JQ1 treatment group, tumor growth was significantly inhibited compared to the vehicle control group (tumor volume: $P = 2 \times 10^{-5}$, tumor weight: $P = 1 \times 10^{-4}$,

respectively, two-tailed Student's *t*-test). Six replicates for each group were used and the median with SD was shown (**C** and **E**). qPCR of E6/E7 is shown between JQ1 treatment and vehicle control. Ten replicates for each group were used and median with SD were shown. E6/E7 expression was also reduced after JQ1 treatment ($P < 1 \times 10^{-4}$, two-tailed Sstudent's *t*-test) (**F**). JQ1 treatment for hybrid ecDNA(−) PDX (PDX_C and PDX004) was also performed. Six replicates for each group were used and the median with SD was shown (**G**–**L**). Neither tumor volume nor tumor weight was inhibited significantly compared to the vehicle control group in PDX_C (**G**–**I**) and PDX004 (**J**–**L**) (two-tailed Student's *t*-test). Source Data is available for panels **C**–**F**, **H**, **I**, and **K**, **L**.

## Whole genome sequencing

DNA was extracted using the QIAquick DNA mini kit (Qiagen, Hilden, Germany) for high-quality extraction per the manufacturer's instructions. Library preparation and sequencing were performed with the Illumina NovaSeq 6000 at the UCSD IGM Genomics Center. Hg38 and HPV genome sequences (accession number: AY686584.1) were used for the reference genome.

## AmpliconArchitect

WGS data were analyzed using AmpliconSuite-pipeline, which encompasses AmpliconArchitect[31] v1.3.r3 and AmpliconClassifier[2] v0.5.1 to detect hybrid ecDNA, using the GRCh38_viral reference genome option (https://github.com/AmpliconSuite). In brief, CNVkit v0.9.9 was employed for copy number segmentation and estimation. Segments with copy number ≥ 2.5 copies above chromosome arm ploidy, as well as viral genome regions with CN ≥ 1 were extracted using the AmpliconSuite-pipeline and used as seed regions for analysis of focal amplifications. For each seed region, AmpliconArchitect searched the region and nearby loci for discordant read pairs, which are indicative of genomic structural rearrangements. Genomic segments are defined based on the positions of gene breakpoints and changes in copy number. AmpliconArchitect utilizes structural variant signatures, such as discordant paired-end reads and CNV boundaries, to partition all intervals into segments. It then constructs an amplicon graph based on these segments and decomposes the graph into genome paths and cycles that explain the observed changes in copy number and structural variation. The Amplicon-Classifier script was used to classify amplicons into categories such as ecDNA, breakage-fusion-bridge, complex non-cyclic, linear, and no focal amplification based on rules related to patterns of structural variation, copy number and decomposed genome paths from AmpliconArchitect. Circular visualizations of ecDNA genome structure and annotation tracks were generated using CycleViz (https://github.com/AmpliconSuite/CycleViz)[32].

## RNA extraction and RNA-seq

RNA was extracted following the protocol of the Qiagen RNeasy Plus Mini Kit (Qiagen, Hilden, Germany). RNA concentration, purity, and integrity were verified using a NanoDrop spectrophotometer (Thermo Fisher Scientific, Waltham, MA). In addition, an RNA Integrity Number (RIN) of 7.0 or greater by TapeStation was required for quality assessment. Library preparation and RNA sequencing were performed by the UCSD IGM Genomics Center utilizing an Illumina NovaSeq 6000.

## Long read DNA-sequencing

High molecular weight DNA(> 40 kb) was extracted using the Nanobind CBB kit or Nanobing tissue kit (PacBio, Menlo Park, CA) following the manufacturer's recommendations. Long-read DNA sequencing was performed using 1 SMRT Cell per sample. Library preparation and sequencing were performed by the UCSD IGM Genomics Center with the Illumina NovaSeq 6000.

## ChIP-seq

For ChIP experiments, $1 \times 10^7$ cells for cell lines, and 25 mg tumors for PDX were used for each condition. Antibody for each immunoprecipitation using H3K4me3 (#9751, 2 μg, Cell Signaling Technology, Danvers, MA), H3K4me1 (#C5326S, 2 μg, Cell Signaling Technology), H3K27ac (#91194, 4 μg, Active Motif, Carlsbad, CA), BRD4 (#A301-985A50, 7.5 μg, Bethyl Laboratories, Montgomery, TX), and IgG (#2729, 2 μg, Cell Signaling Technology) for negative control was used. We used the SimpleChIP Kit (Cell Signaling Technology, Danvers, MA) and followed the manufacturer's protocol. Library preparation and sequencing were performed with the Illumina NovaSeq 6000 at the UC San Diego IGM Genomics Center.

## ATAC-seq

Around 50 mg of pulverized PDX tumor tissue with liquid nitrogen and thawed cell lines ($3 \times 10^6$) were used for nuclei preparation. PDX tumor tissue was dissociated by GentleMACS machine in MACS buffer [Tris−HCl, pH = 8 (Thermo Fisher Scientific), 5 $CaCl_2$ (G-Biosciences, St. Louis, MO), 3 mM Mg-acetate (Sigma-Aldrich), 2 mM EDTA (Thermo Fisher Scientific), 0.6 mM DTT (Sigma-Aldrich) and Protease inhibitor (Roche, Basel, Switzerland) in Molecular biology water (Corning, Corning, NY)]. Nuclei were pelleted by centrifugation for 5 min at 500×$g$ at 4 °C. Permeabilized nuclei were obtained by resuspending nuclei pellet in 1 mL Nuclear Permeabilization Buffer [0.2% IGEPAL-CA630 (Sigma-Aldrich), 1 mM DTT (Sigma-Aldrich), Protease inhibitor (Roche, Basel, Switzerland), 5% BSA (Sigma-Aldrich) in PBS (Thermo Fisher Scientific)], and incubating for 10 min on a rotator at 4 °C. For cell lines, permeabilized nuclei were obtained by resuspending cells in 250 μL Nuclear Permeabilization Buffer and incubating for 5 min on a rotator at 4 °C. Nuclei were then pelleted by centrifugation for 5 min at 500×$g$ at 4 °C. The pellet was resuspended in 25 μL ice-cold Tagmentation Buffer [33 mM Tris−acetate (pH = 7.8) (Thermo Fisher Scientific), 66 mM K-acetate (Sigma-Aldrich), 11 mM Mg-acetate (Sigma-Aldrich), 16% DMF (Merk Millipore, Darmstadt, Germany) in Molecular biology water (Corning, Corning, NY)]. An aliquot was then taken and counted by hemocytometer to determine nuclei concentration. Approximately 50,000 nuclei were resuspended in 20 μL ice-cold Tagmentation Buffer and incubated with 1 μL Tagmentation enzyme (Illumina) at 37 °C for 60 min with shaking at 500 rpm. The tagmentated DNA was purified using MinElute PCR purification kit (Qiagen). The libraries were amplified using NEBNext High-Fidelity 2X PCR Master Mix (NEB, Ipswich, MA) with primer extension at 72 °C for 5 min, denaturation at 98 °C for 30 s, followed by 8 cycles of denaturation at 98 °C for 10 s, annealing at 63 °C for 30 s and extension at 72 °C for 60 s. Amplified libraries were then purified using MinElute PCR purification kit (Qiagen), and two size selection steps were performed using SPRIselect bead (Beckman Coulter, Brea, CA) at 0.55× and 1.5× bead-to-sample volume rations, respectively[33].

Final libraries were quantified using Qubit (Thermo Fisher Scientific) and checked for library size distribution using 4200 TapeStation (Agilent Technologies, Santa Clara, CA). Library preparation and sequencing were performed with the Illumina NovaSeq 6000 at the UCSD IGM Genomics Center.

## ChIP-seq and ATAC-seq analysis

Raw sequence data were first trimmed for sequencing adapters using *fastp*[34], then aligned to the human genome (hg38) augmented by the HPV type 16 genome sequence as an extra chromosome (accession number AY686584.1), using *STAR* aligner[35] with chimeric alignments allowed in the output. Peak calling was done in an unbiased way on an artificial sample created as the union of reads sampled randomly (with equal weight per sample) from all samples of the same type (ATAC-seq, H3K27ac, etc.), using HOMER (Hypergeometric Optimization of Motif EnRichment)[36]. Peak quantitation was done for each sample individually by counting reads aligning to the peak regions determined in the previous step.

## Peak normalization and differential analysis

Raw read numbers in peak regions of all samples of the same type were normalized separately. For the H3K27ac mark, we have two replicates each of HMS001 and PDX_A treated with either vehicle or JQ1 (8 samples total). Normalization proceeds under the assumption that the tallest peaks should be comparable across samples. To this end, we found peak regions in which the raw peak values are among the top 30-percentile for each sample. To this subset of all peak regions, we apply the Relative Log Expression (RLE) normalization method[37]. The normalization factors per sample are then applied to all peak regions. This normalization method is insensitive to potentially different levels of

low-level "background" signals in different samples. Differentially acetylated regions (H3K27ac mark) were identified as follows: Since we have only two biological replicates per condition and $\sim 10^4$ tests to carry out, the standard $t$-test has very low power. Instead, for each peak region and each comparison we calculate a $z$-score, in which the standard deviation in the denominator is modeled as a function of the mean signal across samples being compared. This function is obtained as the *loess* regression fit through the calculated standard deviation data versus mean signal, using all eight samples together as if they were replicates. The approximate $z$-scores in every comparison are then assessed for significance by the empirical Bayes method[38]. The result is a posterior error probability *lfdr* assigned to each peak region in every comparison.

## Finding significantly differentially activated peaks (H3K27ac) within active promoters (H3K4me3) and enhancers (H3K4me1)

We look for the intercept of significantly differentially activated genomic regions (H3K27ac peaks with significance operationally defined as *lfdr* <0.3) with a) peak regions defined by the active promoter mark H3K4me3 or b) by the enhancer mark H3K4me1.

## HiC library preparation and sequencing

HiC experiments were performed based on the protocol of Arima-HiC Kit (Arima Genomics, Carlsbad, CA). Briefly, chromatin obtained from each cell line or PDX tumor was first cross-linked and digested using a restriction enzyme cocktail. The digested ends were labeled with biotinylated nucleotides and ligated to capture the sequence and structure of the genome. The ligated DNA was purified and fragmented, and the enriched biotinylated fragments were subjected to a custom library preparation protocol using an Arima Library preparation module. Sequencing was performed with the Illumina NovaSeq 6000 by Arima Genomics.

## Hi-C data processing

Hi-C data were processed by the runHiC python package (https://pypi. org/project/runHiC/) with a custom reference genome composed of human genome assembly hg38 augmented by the HPV type 16 genome as an extra chromosome. The runHiC package can remove duplicate reads, assign reads to restriction fragments, filter out invalid interaction pairs, and generate binned interaction matrices in *.mcool format. We specified the enzyme name as *Arima* in the filtering step to remove the read pair that maps to the same restriction fragment. We used a bin size as small as 1kbp in the binning step to generate a high-resolution interaction matrix for each chromosome. Given the primary ecDNA structure predicted by AmpliconArchitect (and validated by long reads), we extracted the submatrices from each genomic interval composing the ecDNA and assembled these submatrices into one matrix corresponding to ecDNA according to the order and orientation of the corresponding intervals on the ecDNA. Finally, we applied ICE normalization on the resulting matrix and visualized them in Fig. 3. To identify significant Hi-C interactions from a reassembled matrix $C$, we fit the interactions at each genomic distance on the hybrid ecDNA HMS001 and PDX_A using a Poisson distribution with mean interactions at that genomic distance and computed statistical significance ($p$-value) as the probability of observing at least $C_{ij}$ interactions with the Poisson model. Then we correct all resulting $p$-values for multiple testing using the Benjamini−Hochberg procedure to compute a $q$-value for each pair of bins ($i, j$). Pairs with $q$-value < 0.05 were characterized as significant interactions.

## Metaphase chromosome spreads

Cultured cells were enriched in metaphase by treatment with KaryoMAX Colcemid (Gibco) at 100 ng ml$^{-1}$ overnight. After washing once with PBS, single-cell suspensions were incubated in 75 mM KCl for 15 min at 37 °C. Carnoy's fixative (3:1 methanol:glacial acetic acid) was used for cell fixation and cells were spun down. Cells were washed 3 more times with a fixative solution and dropped onto humidified glass slides.

## Fluorescence in situ hybridization (FISH)

Fixed cells on slides from cell lines or primary cultured cells from PDX were equilibrated in 2xSCC buffer and dehydrated in 70%, 85%, and 100% ethyl alcohol for ~2 min each. FISH probes for human genome parts of hybrid ecDNA (EYA2 and NDUFC1 FISH probe) were purchased from EMPIRE genomics (Depew, NY), and HPV16 probes were purchased from Arbor Biosciences (Ann Arbor, MI). Diluted FISH probes in hybridization buffer were added to the sample and covered with a coverslip. Slides were denatured at 72 °C for 1 min and hybridized overnight at 37 °C. The slides were then washed with 0.4× SSC, and 2× SSC-0.1% Tween 20. DAPI was applied to the samples for 1 min before washing again and mounting with Prolong Gold.

## Microscopy

Confocal images were collected on a Nikon confocal plus STORM system using a 100 ×1.49 NA TIRF objective (Moores Cancer Center Shared Resources). For STORM, samples were mounted in STORM buffer (50 mM Tris, pH 8.0, 10 mM NaCl, 10% glucose, 0.1 M mercaptoethanolamine 56 units/ml glucose oxidase, and 340 units/ml catalase) and duplicate images were collected at the same pixel size (0.16 μm and 256 × 256 field of view) in the confocal mode and in the super-resolution mode using an ANDOR IXON3 Ultra DU897 EMCCD camera to enable overlay of the confocal (DAPI) and STORM images. For super-resolution images, TIRF illumination settings were used as appropriate to enhance the signal-to-noise ratio. The images were collected at frame rate (about 15 ms exposure time using a 256 × 256 pixel area of the camera chip) using the sequential illumination setting in the STORM acquisition module in Nikon Elements software (version 4.6). Laser power was adjusted so that 50−350 localization events were recorded per channel in each 256 × 256 pixel area frame. Acquisition was stopped once 1−3 million localization events were recorded. Analysis of the image stacks was carried out using the STORM analysis module of the Elements software. The STORM images were superimposed on the confocal DAPI images for context.

## CRISPRi

We used doxycycline-inducive dCas9-KRAB plasmid (Addgene, 50917) for CRISPRi experiments[39]. Lentiviral transduction of this plasmid to HMS001 was performed using HEK293T cell (ATCC, CRL-3216) with CRISPR&MISSION Lentiviral Packaging Mix (Sigma-Aldrich). Stable dCas9-expressing cells were confirmed by Western blot for Cas9 (Cell Signaling Technology). Lentiviral transduction was also performed using each target sgRNA. SgRNAs were created to target the enhancers on hybrid ecDNA S1 (#1) and S2 (#2) with non-targeting control (nonT) (GenScript, Piscataway, NJ) (Supplemental Table 2). To confirm each gene expression from transduced cell lines, Western blotting for Cas9 (#14697S, 1:1000, Cell Signaling Technology), MYC (#13987S, 1:1000, Cell Signaling Technology), E6 (#GTX132686, 1:100, GeneTex, Irvine, CA), E7 (#GTX133411, 1:100, GeneTex), and GAPDH (#2118S, 1:6000, Cell Signaling Technology) or quantitative PCR for Cas9 and ACTB were used (Supplementary Table 3). Difference in proliferation was measured as the ratio of the relative absorbance 2 days after the start of measurement between doxycycline(+) vs. doxycycline(−).

## BET inhibitor treatment for cell lines

The effect of BET inhibitor was investigated on cell lines HMS001 and SCC154. Proliferation was investigated in the presence of DMSO (negative control), as well as JQ1 (10 nM, 100 nM, and 1 μM) (SML1524, Sigma-Aldrich). Cells were seeded in 96-well plates at a density of 8000 cells/well for HMS001 and 4000 cells/well for SCC154. Absorbance at 540 and 590 nm were measured at day 0 and 2 days after the JQ1 or DMSO

treatment and relative absorbance day2/day0 was used for tracking the proliferation of each condition. To confirm each gene expression after treatment, Western blotting for MYC (#13987S, 1:1000, Cell Signaling Technology), E6 (#GTX132686, 1:100, GeneTex), E7 (#GTX133411, GeneTex), and GAPDH (#2118S, 1:6000, Cell Signaling Technology) or quantitative PCR for MYC, E6/E7, and ACTB were used (Supplemental Table 3). For proliferation experiments, each datapoint is the average of 5 replicates with SD represented by error bars, and all experiments were repeated at least 3 times demonstrating consistent results.

### BET inhibitor treatment for PDX

When the tumor volume of PDX (PDX_A: 55-year-old male, PDX_C: 76-year-old male, and PDX004: 75-year-old male) reached about 150–250 mm³, these mice were randomly divided into 2 groups (each group, $n = 6$). Mice were treated daily with either JQ1 at 50 mg/kg IP or vehicle control. Tumor samples were harvested after 2 weeks of JQ1 treatment. To confirm each gene expression after treatment, quantitative PCR for E6/E7 and ACTB was performed using tumors of PDX_A (Supplemental Table 2)

### Gene Ontology (GO) analysis

Gene annotation enrichment analysis was conducted using the Functional Annotation tool of Metascape (http://metascape.org/gp/index.html#/main/step1)[40].

### Statistics and reproducibility

Comparison of the results of qPCR, proliferation assay, PDX tumor volume, and PDX tumor weight in each group was analyzed by two-tailed Student's $t$-test using GraphPad Prism version 10.1. Every experiment was at least repeated twice with similar results. The sample size for each experiment was determined based on pilot experiments and a review of the literature. Samples were allocated randomly in each experimental group.

### Reporting summary

Further information on research design is available in the Nature Portfolio Reporting Summary linked to this article.

## Data availability

Sequencing data of each cell line and PDX tumor data have been deposited in Sequence Read Archive (SRA) database under accession code PRJNA1200897 at "https://www.ncbi.nlm.nih.gov/bioproject/PRJNA1200897/". AmpliconArchitect outputs associated with this study can be found at "https://ampliconrepository.org/project/667314504677da1a8e41a4c0". Source data are provided with this paper.

## Code availability

Code data for this study are available at: https://github.com/virajbdeshpande/AmpliconArchitect, https://github.com/AmpliconSuite/CycleViz.

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

## Acknowledgements

We thank Cynthia Fox for English language editing. Funding includes support from the Gleiberman Head and Neck Cancer Center, Japan Society for the Promotion of Science Overseas Research Fellowship, and National Institutes of Health, Grants 2UL1TR001442-08 of CTSA.

## Author contributions

T.N.: Conceptualization, data curation, formal analysis, investigation, visualization, methodology, writing—original draft, writing—review, and editing. J.L.: Software, bioinformatic analysis, investigation, visualization, methodology, writing–review and editing. K.Z.: Software, bioinformatic analysis, investigation, visualization, methodology, writing—review and editing. J.T.L.: Methodology. R.S.: Data curation, visualization, methodology. C.P.: Data curation, investigation. S.S.: Investigation. S.J.: Data curation, investigation. Q.Y.: Data curation, investigation. A.M.: Investigation. S.F.: Data curation. A.W.: writing—review and editing. K.P.: visualization, methodology. B.R.: Writing—review and editing. K.M.F.: Writing—review and editing. P.M.: Writing—review and editing. V.B.: Conceptualization, resources, supervision, funding acquisition, investigation, writing—original draft, writing–review and editing. J.A.C: Conceptualization, formal analysis, resources, supervision, funding acquisition, investigation, writing—original draft, project administration, writing—review and editing.

## Competing interests

P.M. is a co-founder, chairs the scientific advisory board (SAB) of, and has equity interest in Boundless Bio Inc. (BBI). P.M. is also an advisor with equity for Asteroid Therapeutics and is an advisor to Sage Therapeutics. V.B. is a co-founder, consultant, SAB member and has equity interest in Boundless Bio and Abterra. J.T.L. is an employee of BBI. J.L. previously consulted for BBI. The other authors have no competing interests to disclose.
