## [Peer Review file · Nature Communications]

Inhibition of human-HPV hybrid ecDNA enhancers reduces oncogene expression and tumor growth in oropharyngeal cancer

Corresponding Author: Professor Joseph Califano

Version 0:

Reviewer comments:

Reviewer #1

(Remarks to the Author)

The Nakagawa manuscript describes the characterization of HPV-positive HNSCC cell lines and xenografts, an analysis of ecDNA, epigenetic status, and a preliminary therapeutic strategy targeting ecDNA-positive tumors. The author presents data suggesting that new enhancers in cellular or viral DNA are created in ecDNA.

The manuscript is well-written, and the methods and data are presented clearly. The paper could be improved as follows:

Major

-A limitations section of the discussion is needed, pointing out that the sample size is extremely small and, therefore, results are very preliminary

-The suggestion that SCC154 contains HPV-only episomes is not convincing if the only data is the very faint FISH signal in Fig 1g. Is there any confirmatory data from long-read sequencing?

-If JQ1 inhibits MYC, is this not likely to reduce proliferation even in ecDNA- cells?

The description of the enhancer identified in the HPV L1 gene region needs to be more precisely defined. Can the authors provide a clear boundary (it is really hard to see in Figure 2B)? In particular, this needs to be distinguished from the known enhancer in the HPV URR region, which is not mentioned.

-Is it not expected that all the ecDNA would display as open/accessible chromatin by ATAC sequencing?

I am not an expert on Hi-C analysis, but it seems there is a lot of background, especially in Figure 3A, where HPV interacts with nearly all regions of the human sequence in the ecDNA. Are the interactions pointed to by the arrows significantly higher?

-Does long-read sequencing support the presence of episomal HPV DNA in HMS001?

Minor

L 191-4. It is confusing that JQ1 is the only BRD inhibitor used. We focused using the Bromo- and Extra- Terminal domain (BET) inhibitors as potential therapeutic agents that specifically target BRD4 as a key linker of ecDNA6.

Suggested to read "We tested a Bromo- and Extra-Terminal domain (BET) inhibitor as a potential therapeutic agent to target BRD4 as a key linker of ecDNA6."

L 717 yellow allow should be arrow

Fig. 2: Detecting active enhancer using ChIP-seq, and new identification of HPV integration mechanism in hybrid ecDNA. This title needs to be clarified. Do you mean "Fig. 2: Detecting active enhancers using ChIP-seq, and identifying new HPV integration mechanisms in hybrid ecDNA"?

L 720 an expanded should be an expanded.

L781 JQ1 treatment on should be "JQ1 treatment of". Please proofread all the figure legends; every sentence of the Figure 6 legend is awkward.

Michael Dean

Reviewer #2

(Remarks to the Author)

The authors of the submitted article, "Inhibition of novel human-HPV hybrid ecDNA enhancers reduces oncogene expression and tumor growth in oropharyngeal cancer," present a tour de force for sequencing and analysis providing firm evidence of a new mechanism driving high levels of HPV oncogene expression, namely hybrid ecDNA with creation of new enhancers from HPV and somatic sequences. Sequencing techniques including ChIP, ATAC, Long read, and HiC were complemented by strong informatics and findings of hybrid ecDNA and novel enhancers confirmed by FISH, CRISPR, and BET inhibitor studies in a cell line and PDX containing hybrid ecDNA. The studies expand our understanding of HPV oncogene expression and raise potential for therapies targeting a subset of tumors.

A few issues should be clarified or expanded.

Methods for proliferation assays state differences in absorbance, but it is unclear absorbance of what.

In the manuscript, the first time that JQ1 is mentioned it should be identified as a BET inhibitor (line 198).

Figure 4E, proliferation with targeting S1 for HMS001 should be shown as an additional control

Figure 5 and line 199 stating signals were significantly reduced after JQ1 treatment, how was this determined? How many cells were analyzed and what were differences with and without JQ1 treatment?

Figure 7, is E needed? Seems that E is the same data as D

Figure 7 should show E6/7 expression in PDXC and PDX004 as was shown for PDXA.

These studies expand our understanding and raise many interesting questions about response of tumors with and without hybrid ecDNA-driven expression of HPV oncogenes such as: is there a differential response to standard or DNA damaging therapy, what are molecular characteristics required for cellular maintenance of ecDNA and do these characteristics differ from those required for maintenance of HPV episomes, and are there other potential ways to target these cells?

Reviewer #3

(Remarks to the Author)

In this manuscript, Nakagawa and colleagues expand upon an important theme in HPV cancer research concerning the nature and functional consequences of extrachromosomal DNA in HPV-positive cancer cells. As several laboratories have demonstrated, HPV-positive cancer cells exhibit significant chromosomal instability and have recently been shown to often harbor extrachromosomal DNA (ecDNA) molecules that harbor HPV-human hybrid sequences. Selective pressure to maintain expression of the HPV E6 and E7 oncogenes in cancer cells likely leads to the maintenance of such ecDNA molecules. However, the consequences of ecDNA maintenance with respect to gene expression and cancer cell viability have been poorly understood. The authors therefore characterize two HPV-positive cancer cell lines and two HPV-positive patient derived xenografts, one of each of which harbors detectable ecDNA. They also compare these to an HPV-negative spontaneously immortalized human cell line not known to harbor ecDNA. They determine that the hybrid ecDNAs contain active enhancers that share common features even though they differ in sequence, and that treatment of the ecDNA-containing cells with the BRD4 inhibitor JQ1 limits the growth of those cells. In general, the experiments are carefully executed and the data are clearly presented. Some of the predicted links between enhancer characterization and gene expression seem somewhat premature, and the manuscript would benefit from additional explanation of certain conclusions.

Specific comments:

1. How is it determined that certain cell lines do not contain any ecDNA?
2. In 1G, the red signal is difficult to appreciate. Can the region of interest be enlarged?
3. In 3B, what is the nature of the small region of sequence (denoted in grey) between the two enhancer regions?
4. Can the authors clarify whether there is enhanced expression of any of the cellular sequences present in the ecDNA molecules, compared to controls?
5. Major point: the authors conduct extensive ChIP-seq experiments and define active enhancers in the ecDNA-containing cancer cell lines. They then conduct gene ontology analyses to infer changes in gene expression based on these enhancer marks.
 - a. However, enhancer-promoter interactions are now understood to take place over long distances in the cellular genome, and there is no explanation of how an active enhancer was mapped to a particular gene. If the enhancer was assigned only to its most proximal promoter, this approach may not take advantage of current information about enhancer-promoter interactions and the control of gene expression.
 - b. Furthermore, the changes in gene expression that are predicted from the results of these enhancer analyses should be tested experimentally.
6. The statement on lines 269-270 "In this case, the S1 enhancer was working the enhancer complex with L1 enhancer of HPV." is unclear.

Version 1:

Reviewer comments:

Reviewer #1

(Remarks to the Author)

Thank you for clear and concise responses to the questions raised by myself and the other reviewers.

Michael Dean

(Remarks on code availability)

Reviewer #2

(Remarks to the Author)

The authors have adequately addressed concerns of the reviewer.

(Remarks on code availability)

Reviewer #3

(Remarks to the Author)

Thank you for addressing my questions and comments, and especially for the inclusion of new RNA-seq data.

(Remarks on code availability)

Dear Reviewers:

Thank you for your letter and for the reviewers' comments concerning our manuscript entitled "Inhibition of novel human-HPV hybrid ecDNA enhancers reduces oncogene expression and tumor growth in oropharyngeal cancer" as an article for *Nature Communications*. We are very grateful to the editors and reviewers for their contribution to the review. Their comments were very helpful in revising and improving our paper, as well as important guiding implications for our study. We modified our manuscript according to their comments. Accordingly, we have uploaded the revised manuscript with all the changes highlighted by red characters. Appended to this letter is our point-by-point response to the comments raised by the reviewers and editors.

REVIEWER COMMENTS

Reviewer #1 (Remarks to the Author):

The Nakagawa manuscript describes the characterization of HPV-positive HNSCC cell lines and xenografts, an analysis of ecDNA, epigenetic status, and a preliminary therapeutic strategy targeting ecDNA-positive tumors. The author presents data suggesting that new enhancers in cellular or viral DNA are created in ecDNA.

The manuscript is well-written, and the methods and data are presented clearly. The paper could be improved as follows:

Major

-A limitations section of the discussion is needed, pointing out that the sample size is extremely small and, therefore, results are very preliminary

Response. We appreciate this point. We have included section describing limitations in the discussion section in line 295-297 and pointed the sample size is small and acknowledge the impact on results and generalizability.

-The suggestion that SCC154 contains HPV-only episomes is not convincing if the only data is the very faint FISH signal in Fig 1g. Is there any confirmatory data from long-read sequencing?

Response. We acknowledge this comment. Although we did not perform long read sequencing for SCC154, to further analyze the status of HPV in SCC154, we obtained the bam files of Oxford Nanopore long read sequencing data¹, from the authors.

Interestingly, the long read sequencing for SCC154 only shows evidence of integrated HPV into chromosome 21. We realigned all reads over 6kbp in length to the HPV genome using minimap2 and identified 25 reads having mappings to the viral genome. We then examined the alignment of those reads against the original combined reference of the human genome and HPV16, and found that all but one of those reads had combined mapping joining HPV to the human reference between chr21:15211092-15315692, and the remaining read fully mapped better to chr8 than viral sequence.

This finding is consistent with a previous report about the integration status of HPV into chromosomal DNA in SCC154^{1,2}.

Furthermore, the Amplicon Architect result shows CN=2 for HPV and has an SV edge exiting the viral episome but which has an unmapped destination (blue vertical line). This is probably

the integration point on chr21 which we originally could not fully detect with short reads.

Based on the analysis above, The HPV-only signals in chromosome in SCC154 were integrated HPV signals in human chromosome.

Accordingly, we modified L125-127, L133-134 and Fig.1G.

-If JQ1 inhibits MYC, is this not likely to reduce proliferation even in ecDNA- cells?

The description of the enhancer identified in the HPV L1 gene region needs to be more precisely defined. Can the authors provide a clear boundary (it is really hard to see in Figure 2B)? In particular, this needs to be distinguished from the known enhancer in the HPV URR region, which is not mentioned.

Response. As the reviewer mentioned, ecDNA(-) cells (SCC154) also had a slight trend of inhibition of proliferation in response to JQ1, as shown in Fig. 6D. However, while ecDNA(+) cell line showed the significant inhibition of the proliferation, ecDNA(-) cell line showed no significant inhibition. This suggests that MYC inhibition alone was not enough to inhibit the proliferation.

In addition, we show the clear boundary figures of HMS001 and PDX_A like below.

Each of enhancer peak was located on the L1 region that is distinct from the known enhancer in the HPV URR region.

HMS001

PDX_A

-Is it not expected that all the ecDNA would display as open/accessible chromatin by ATAC sequencing?

Response. Since the peaks on the HPV are very high in ATAC-seq results in HMS001, entire peaks on human genome in hybrid ecDNA look very low. However, if we compare entire peaks on human genome in hybrid ecDNA in HMS001 with other cell lines, they display open chromatin compared to the same regions in other cell line that does not have hybrid ecDNA as below (*left*). The results were similar in PDX_A like shown below (*right*).

-I am not an expert on Hi-C analysis, but it seems there is a lot of background, especially in Figure 3A, where HPV interacts with nearly all regions of the human sequence in the ecDNA. Are the interactions pointed to by the arrows significantly higher?

Response. We agree with the reviewer that indeed HPV interacts with nearly all chromosomal segments in the hybrid ecDNA. Therefore, we checked the significant interaction in 1K resolution between human genome and HPV genome in each of hybrid ecDNA, and blue dots were added as indicators of significant interaction in Fig. 3A and D, respectively.

-Does long-read sequencing support the presence of episomal HPV DNA in HMS001?

Response. We appreciate the reviewer raising this important point. To answer this query, we performed additional analysis of our PacBio long read data and employed a similar analysis strategy to the analysis performed to detect HPV episome status for SCC154. In HMS001 we only observed evidence of hybrid human-viral ecDNA, and not HPV16-only episomes.

We first mapped our PacBio reads longer than 5000bp to the HPV16 genome with minimap2 and identified 39 reads with alignments to HPV. Importantly, in a manner completely consistent with the paired-end WGS data (Fig. 1A), in the PacBio data, the region between sequence HPV16:1600-6689 was fully deleted, suggesting that no fully intact viral episomes exist in the sample.

HMS001 PacBio data HPV16 alignments:

While this shows that no full-length HPV-only episomes are supported by the PacBio data, it does not on its own preclude a more exotic hypothesis that the truncated HPV episomes with the same deletion as the human-viral DNA may exist. We examined the sequence alignments of the HMS001 reads to the HPV16 genome and found no reads that suggested a structure where the two endpoints of HPV16 used for integration into the human genome had become fused together without human DNA intervening.

We also analyzed PDX_A to see if it suggested additional copies of HPV-only episomes in addition to the hybrid human-viral ecDNA. We aligned the PDX_A reads to the HPV16 genome with minimap2, and identified 43 reads with alignments to the HPV genome. Similarly to HMS001, we found a complete deletion between the integration points, suggesting that integration into the human genome is a mechanism for creating deletions in the HPV16 genome. Again, like HMS001 we also found no alignments that joined the two ends of the integration points to suggest a shortened HPV-only episome. We added Supplementally fig.1C and D, and L120-124.

Instead the most likely explanation is that only human-viral HPV sequence exists in HMS001 and PDX_A. These data suggest that extrachromosomal HPV-only signals in multi-FISH of HMS001 and PDX_A might be hybrid ecDNAs that did not show sufficient green signals due to difference in the accuracy of each probe.

PDX_A PacBio data HPV16 alignments:

Minor

L 191-4. It is confusing that JQ1 is the only BRD inhibitor used. We focused using the Bromo- and Extra- Terminal domain (BET) inhibitors as potential therapeutic agents that specifically target BRD4 as a key linker of ecDNA6.

Suggested to read “We tested a Bromo- and Extra-Terminal domain (BET) inhibitor as a potential therapeutic agent to target BRD4 as a key linker of ecDNA6.”

Response. We appreciate the reviewer’s great suggestion. We modified the sentence as the reviewer suggested; “We tested a Bromo- and Extra-Terminal domain (BET) inhibitor (JQ1) as a potential therapeutic agent to target BRD4 as a key linker of ecDNA.” in L199-200.

L 717 yellow allow should be arrow

Response. We modified it as the reviewer suggested (yellow allow in L742→ allow, and white allowhead in L744→ allowhead).

Fig. 2: Detecting active enhancer using ChIP-seq, and new identification of HPV integration mechanism in hybrid ecDNA. This title needs to be clarified. Do you mean "Fig. 2: Detecting active enhancers using ChIP-seq, and identifying new HPV integration mechanisms in hybrid ecDNA"?

Response. We modified it, “Detecting active enhancers using ChIP-seq, and identifying new HPV integration mechanisms in hybrid ecDNA.”, as the reviewer suggested.

L 720 anexpanded should be an expanded.

L781 JQ1 treatment on should be “ JQ1 treatment of”. Please proofread all the figure legends; every sentence of the Figure 6 legend is awkward.

Michael Dean

Response. We have made modification as the reviewer suggested. We also proofread all the figure legends as the reviewer suggested.

Reviewer #2 (Remarks to the Author):

The authors of the submitted article, “Inhibition of novel human-HPV hybrid ecDNA enhancers reduces oncogene expression and tumor growth in oropharyngeal cancer,” present a tour de force for sequencing and analysis providing firm evidence of a new mechanism driving high levels of HPV oncogene expression, namely hybrid ecDNA with creation of new enhancers from HPV and somatic sequences. Sequencing techniques including ChIP, ATAC, Long read, and HiC were complemented by strong informatics and findings of hybrid ecDNA and novel enhancers confirmed by FISH, CRISPR, and BET inhibitor studies in a cell line and PDX containing hybrid ecDNA. The studies expand our understanding of HPV oncogene expression and raise potential for therapies targeting a subset of tumors.

A few issues should be clarified or expanded.

Methods for proliferation assays state differences in absorbance, but it is unclear absorbance of what.

In the manuscript, the first time that JQ1 is mentioned it should be identified as a BET inhibitor (line 198).

Response. We appreciate the reviewer's comment. We modified this text as the reviewer suggested in line 580-582; "Absorbance at 540nm and 590nm were measured at day 0 and 2 days after the JQ1 or DMSO treatment and relative absorbance day2/day0 was used for tracking the proliferation of each condition."

We modified the sentences in line 199-200 as the reviewer suggested; "We tested a Bromo- and Extra-Terminal domain (BET) inhibitor (JQ1) as a potential therapeutic agent to target BRD4 as a key linker of ecDNA."

Figure 4E, proliferation with targeting S1 for HMS001 should be shown as an additional control

Response. We agree with the reviewer. We added the proliferation results targeting S1 for HMS001 in Fig. 4E, F, and G as an additional control.

Figure 5 and line 199 stating signals were significantly reduced after JQ1 treatment, how was this determined? How many cells were analyzed and what were differences with and without JQ1 treatment?

Response. We appreciate the reviewer raising important point. Dozens of cells can be seen in one field of view, of which only a few have hybrid ecDNA. We are targeting among them, so we should not use "significant", as the reviewer pointed out. This was caused by the low copy number of hybrid ecDNA compared to other cancer types. We have corrected the statement as the reviewer suggested.

Figure 7, is E needed? Seems that E is the same data as D

Response. We agree with the reviewer. We have deleted Figure 7E as the reviewer suggested.

Figure 7 should show E6/7 expression in PDXC and PDX004 as was shown for PDXA.

Response. We agree with the reviewer. Unfortunately, we performed the qPCR only for PDX_A that showed the significant inhibition of proliferation. However, in cell line, we confirmed the

E6/E7 expression of SCC154 (hybrid ecDNA(-)) was not inhibited significantly after JQ1 treatment (S Fig.3). This result supports the reviewer's comment.

These studies expand our understanding and raise many interesting questions about response of tumors with and without hybrid ecDNA-driven expression of HPV oncogenes such as: is there a differential response to standard or DNA damaging therapy, what are molecular characteristics required for cellular maintenance of ecDNA and do these characteristics differ from those required for maintenance of HPV episomes, and are there other potential ways to target these cells?

Response. We appreciate the reviewer's insightful comments. We are going to continue the research of hybrid ecDNA and want to elucidate the additional targeted therapies based on multidisciplinary molecular mechanisms, as the reviewer mentioned.

Reviewer #3 (Remarks to the Author):

In this manuscript, Nakagawa and colleagues expand upon an important theme in HPV cancer research concerning the nature and functional consequences of extrachromosomal DNA in HPV-positive cancer cells. As several laboratories have demonstrated, HPV-positive cancer cells exhibit significant chromosomal instability and have recently been shown to often harbor extrachromosomal DNA (ecDNA) molecules that harbor HPV-human hybrid sequences. Selective pressure to maintain expression of the HPV E6 and E7 oncogenes in cancer cells likely leads to the maintenance of such ecDNA molecules. However, the consequences of ecDNA maintenance with respect to gene expression and cancer cell viability have been poorly understood. The authors therefore characterize two HPV-positive cancer cell lines and two HPV-positive patient derived xenografts, one of each of which harbors detectable ecDNA. They also compare these to an HPV-negative spontaneously immortalized human cell line not known to harbor ecDNA. They determine that the hybrid ecDNAs contain active enhancers that share common features even though they differ in sequence, and that treatment of the ecDNA-containing cells with the BRD4 inhibitor JQ1 limits the growth of those cells. In general, the experiments are carefully executed and the data are clearly presented. Some of the predicted

links between enhancer characterization and gene expression seem somewhat premature, and the manuscript would benefit from additional explanation of certain conclusions.

Specific comments:

1. How is it determined that certain cell lines do not contain any ecDNA?

Response. We used the AmpliconSuite-pipeline workflow for ecDNA detection which incorporates the AmpliconArchitect method for ecDNA detection. To give a broad summary of this approach, we first called genome-wide copy numbers with CNVkit, including for viral genomes (e.g. HPV16, HPV35, etc.). This information is used by the AmpliconArchitect tool to decide where it should begin the search for focal amplifications like ecDNA. The bioinformatic search for human-viral ecDNA and human-only ecDNA are slightly different. The viral genome is included in the analysis of ecDNA if its copy number is found to be 1 or greater in the genome. Human segments are included if they have 2.5 copies in excess of the median chromosome arm ploidy they are derived from.

AmpliconArchitect uses a “seed and extend” approach whereby it iteratively searches the genome for structural variants connected to amplified genome segments (which are sometimes not initially identified during the coarse-grained seeding stage described above). This enables very sensitive discovery of ecDNA even if a segment is not initially detected to be amplified with coarse-grained whole genome copy number calling.

It is of course true that very rare ecDNA below the copy number threshold would be missed, however we argue that the thresholding used here is aligned with the natural limitations of WGS to accurately detect such events. For reference, NOKSI, which is known to be HPV-negative using highly sensitive approaches like PCR, has a CNVkit estimated HPV16 CN of 0.000545, which is far below the threshold we require for detection of HPV. The small non-zero value is likely due to a handful of mapping artifacts from sequencing.

Some co-authors of this manuscript are developers of the AmpliconSuite methods used in this work to identify ecDNA, (Luebeck et al., bioRxiv 2024.

<https://www.biorxiv.org/content/10.1101/2024.05.06.592768v1>). In benchmarking of human-only ecDNA in cancer cell line data with cytogenetically validated ecDNA status, Luebeck et al.,

reports 90% sensitivity for discovery of ecDNA and a specificity of 78%, indicating robust but not perfect performance for detecting ecDNA (see Fig. 2c of Luebeck, et al., reproduced below).

2. In 1G, the red signal is difficult to appreciate. Can the region of interest be enlarged?

Response. We agree with the reviewer's comment. An expanded view of each signal is additionally shown at upper right corner in Fig. 1E-H.

3. In 3B, what is the nature of the small region of sequence (denoted in grey) between the two enhancer regions?

Response. We thank the reviewer for asking this clarifying question as we agree it is not immediately obvious. This gray region is an indicator that a structural variant connects the two genome sequences being displayed. This way we are able to show the ordered and oriented genome structure of the ecDNA. We have included below a diagram that illustrates how SVs are encoded into the CycleViz figures. The same explanation of SV representation applies for for Fig. 3B.

4. Can the authors clarify whether there is enhanced expression of any of the cellular sequences present in the ecDNA molecules, compared to controls?

Response.

To demonstrate increased expression on the ecDNA, we compared Illumina RNA-seq data from two cell lines HMS001 and PDX_A. We mapped the reads to a GRCh38 reference genome using the STAR aligner with default settings (version=2.7.11b). In HMS001, the human region chr20:47028055-47058058 is amplified on ecDNA. In PDX_A on the other hand, the human region chr4:139193557-139444466 is amplified on ecDNA (figures were shown below). Otherwise the two cell lines are well matched as each is an oropharyngeal cancer cell line positive for hybrid human-viral ecDNA.

When we compared the relative expression of RNA along the human sequences in both cell lines, we found massive increases in expression (average coverage per base normalized by the number of reads in each sample)

Region chr4:139193557-139444466 (PDX-A hybrid ecDNA):

HMS001 coverage: 0.22 RPM

PDX-A coverage: 3.15 RPM

PDX-A/HMS001 ratio: 14.64x

Region chr20:47028055-47058058 (HMS001 hybrid ecDNA):

HMS001 coverage: 2.38 RPM

PDX-A coverage: 0.02 RPM

HMS001/PDX-A ratio: 109.28x

These indicate a high-degree of additional expression due to the hybrid ecDNA. The finding that ecDNA contributes to high gene (and surrounding genome sequence) expression is consistent with many prior reports³⁻⁶. We added L153-155, and modified Fig. 2B and D.

5. Major point: the authors conduct extensive ChIP-seq experiments and define active enhancers in the ecDNA-containing cancer cell lines. They then conduct gene ontology analyses to infer changes in gene expression based on these enhancer marks.

a. However, enhancer-promoter interactions are now understood to take place over long distances in the cellular genome, and there is no explanation of how an active enhancer was mapped to a particular gene. If the enhancer was assigned only to its most proximal promoter, this approach

may not take advantage of current information about enhancer-promoter interactions and the control of gene expression.

Response. We appreciate this insightful comment and agree with the reviewer. The fact that enhancers can act at a distance is of course, correct. As the reviewer mentioned, active enhancers were assigned only to the most proximal promoters in this study. Even though this is the typical way to check the enhancer-promoter interaction, this will not characterize every specific potential Enhancer-Promoter pair. However, the aim of this experiment was to perform functional annotation of groups of upregulated genes that are enriched after JQ1 treatment, and we have identified some of the genes activated. More extensive analysis should be performed using Hi-ChIP to check the Enhancer-Promoter interaction in detail in future study.

b. Furthermore, the changes in gene expression that are predicted from the results of these enhancer analyses should be tested experimentally.

Response. We agree with the reviewer. We performed RNA-seq for JQ1 treatment (+/-) condition in HMS001 additionally, as the reviewer mentioned. Differential gene expression analysis using the RNA-seq data of JQ1 treatment (+/-) showed that genes related with GO term of “Cell cycles” were significantly downregulated, and “positive regulation of programmed cell death” including “apoptosis”, and “p53 transcriptional gene network” were significantly upregulated, consistent with the changes associated with the downregulation of HPV E6/E7 after JQ1 treatment. In addition, the GO terms associated with “apoptosis” of genes up-regulated in these RNA-seq were consistent with the GO terms enriched for active enhancer in the ChIP-seq data, suggesting the involvement in the expression of active enhancers. We added the results in L227-232, and Supplementary Fig. 4D and E.

6. The statement on lines 269-270 “In this case, the S1 enhancer was working the enhancer complex with L1 enhancer of HPV.” is unclear.

Response. I appreciate the reviewer’s comment. We wanted to state here that the S1 and L1 enhancers form one large enhancer together, and repression of S1 enhancer by CRISPRi was not sufficient to repress E6/E7 expression, because L1 enhancer is still active.

We modified the sentences in line 275-278 as the reviewer suggested; “On the other hand, the S1 and L1 enhancers form one large enhancer together, and repression of S1 enhancer alone by CRISPRi was not sufficient to repress E6/E7 expression, because L1 enhancer was still active”.

** See Nature Portfolio’s author and referees' website at www.nature.com/authors for information about policies, services and author benefits.

- 1 Rodriguez, I. *et al.* Insights into the mechanisms and structure of breakage-fusion-bridge cycles in cervical cancer using long-read sequencing. *Am J Hum Genet* **111**, 544-561 (2024). <https://doi.org/10.1016/j.ajhg.2024.01.002>
- 2 Walline, H. M. *et al.* Integration of high-risk human papillomavirus into cellular cancer-related genes in head and neck cancer cell lines. *Head Neck* **39**, 840-852 (2017). <https://doi.org/10.1002/hed.24729>
- 3 Morton, A. R. *et al.* Functional Enhancers Shape Extrachromosomal Oncogene Amplifications. *Cell* **179**, 1330-1341 e1313 (2019). <https://doi.org/10.1016/j.cell.2019.10.039>
- 4 Wu, S. *et al.* Circular ecDNA promotes accessible chromatin and high oncogene expression. *Nature* **575**, 699-703 (2019). <https://doi.org/10.1038/s41586-019-1763-5>
- 5 Pang, J. *et al.* Extrachromosomal DNA in HPV-Mediated Oropharyngeal Cancer Drives Diverse Oncogene Transcription. *Clin Cancer Res* **27**, 6772-6786 (2021). <https://doi.org/10.1158/1078-0432.CCR-21-2484>
- 6 Hung, K. L. *et al.* ecDNA hubs drive cooperative intermolecular oncogene expression. *Nature* **600**, 731-736 (2021). <https://doi.org/10.1038/s41586-021-04116-8>